# Genome-Wide Analysis of Alternative Splicing and Non-Coding RNAs Reveal Complicated Transcriptional Regulation in *Cannabis sativa* L.

**DOI:** 10.3390/ijms222111989

**Published:** 2021-11-05

**Authors:** Bin Wu, Yanni Li, Jishuang Li, Zhenzhen Xie, Mingbao Luan, Chunsheng Gao, Yuhua Shi, Shilin Chen

**Affiliations:** 1Institute of Medicinal Plant Development, Chinese Academy of Medical Sciences and Peking Union Medical College, Beijing 100193, China; bwu@implad.ac.cn (B.W.); liyanni_hua1126@126.com (Y.L.); ljs1378905466@126.com (J.L.); zhzhxie126@126.com (Z.X.); 2Institute of Bast Fiber Crops, Chinese Academy of Agricultural Sciences, Changsha 410205, China; luanmingbao2002@126.com (M.L.); csgao06@163.com (C.G.); 3Institute of Chinese Materia Medica, China Academy of Chinese Medical Sciences, Beijing 100700, China; yhshi@icmm.ac.cn

**Keywords:** *Cannabis sativa* L., alternative splicing, non-coding RNAs, gene expression and regulation

## Abstract

It is of significance to mine the structural genes related to the biosynthetic pathway of fatty acid (FA) and cellulose as well as explore the regulatory mechanism of alternative splicing (AS), microRNAs (miRNAs) and long non-coding RNAs (lncRNAs) in the biosynthesis of cannabinoids, FA and cellulose, which would enhance the knowledge of gene expression and regulation at post-transcriptional level in *Cannabis sativa* L. In this study, transcriptome, small RNA and degradome libraries of hemp ‘Yunma No.1’ were established, and comprehensive analysis was performed. As a result, a total of 154, 32 and 331 transcripts encoding key enzymes involved in the biosynthesis of cannabinoids, FA and cellulose were predicted, respectively, among which AS occurred in 368 transcripts. Moreover, 183 conserved miRNAs, 380 *C. sativa*-specific miRNAs and 7783 lncRNAs were predicted. Among them, 70 miRNAs and 17 lncRNAs potentially targeted 13 and 17 transcripts, respectively, encoding key enzymes or transporters involved in the biosynthesis of cannabinoids, cellulose or FA. Finally, the crosstalk between AS and miRNAs or lncRNAs involved in cannabinoids and cellulose was also predicted. In summary, all these results provided insights into the complicated network of gene expression and regulation in *C. sativa*.

## 1. Introduction

Recently, *Cannabis sativa* L. has attracted worldwide attention due to its wide use in medicine, food, health products, textiles and other fields [1,2]. The main products of *C. sativa* are derived from cannabinoids, fatty acid (FA) and cellulose [3,4,5]. Cannabinoids are types of *C. sativa*-specific secondary metabolites rich in female flowers [6,7]. To date, more than 100 cannabinoid compounds have been found in *C. sativa*, of which Δ9-tetrahydrocannabinolic acid (THCA) and cannabidiolic acid (CBDA) are the major components [8]. Both THCA and CBDA have functions in relieving pain, keeping calm, preventing vomiting, regulating immunity and reducing artery occlusion [9,10]. However, THCA has hallucinogenic and addictive effects [11,12]. Content and composition of cannabinoids are highly variable among *C. sativa* plants. According to the content and ratio of THCA and CBDA, *C. sativa* can be mainly divided into marijuana and hemp. Marijuana contains high THCA and low CBDA, while hemp is the opposite. Besides CBDA, hemp is cultivated for fiber and oil. Hemp seed is a kind of nutritious food, rich in FA [13]. Moreover, the proportion of ω-6 and ω-3 FAs is 3:1, consistent with the recommended balance for human health [14]. The cultivation area and the export of products of hemp in China account for about 50% and 25% of the world, respectively [15].

With the release of the draft genome of *C. sativa*, the biosynthetic pathway of cannabinoids has been deciphered and is involved in the pathways of hexanoate, plastid-localized 2-C-methyl-d-erythritol 4-phosphate (MEP) and geranyl pyrophosphate (GPP) [16,17,18]. The biosynthetic pathway of FA has been studied in some plants [19,20], but there is no report on *C. sativa* so far. The biosynthesis of cellulose is very complex in plants. Besides cellulose synthase (CESA), some non-CESA proteins are also involved in the process [21]. However, the genes related to the biosynthesis of cellulose in *C. sativa* have not been mined yet. In brief, the regulation in the biosynthesis of cannabinoids, FA and cellulose in *C. sativa* is still unknown to date.

It is known that gene expression and regulation consist of many layers. Alternative splicing (AS) is a significant mechanism for regulating the expression of genes in eukaryotes [22]. AS refers to one mRNA precursor producing diverse transcripts, which is thought of as an important post-transcriptional mechanism for increasing the diversity of transcripts and proteins [23]. AS plays a major role in affecting protein function; therefore, it is necessary to systematically identify AS events. However, AS has not been reported in *C. sativa.*

Non-coding RNAs, especially for microRNAs (miRNAs) and long non-coding RNAs (lncRNAs), have been comprehensively studied using bioinformatic and experimental approaches in plants [24,25]. However, they are not reported in *C. sativa*. MiRNAs are types of 20–24 nucleotide non-coding RNAs, usually originating from single-stranded transcripts with imperfect stem-loop structure [26], which cleavage or inhibit translation of target genes at posttranscriptional level or through methylation modification at the transcriptional level [27,28,29]. MiRNAs function as gene regulators during the plant growth process [30,31,32] and are also involved in adaptive responses to abiotic stresses [33,34] and synthesis of secondary metabolites [35,36,37]. LncRNAs have become a popular research topic in molecular biology in recent years. LncRNAs display a key regulatory role in plant cell development, which can be trans-acting for target genes in a sequence complementary manner [38] or cis-acting in other mechanisms [39].

It was reported that AS and miRNAs could crosstalk, which increased the complexity of gene expression and regulation, emerging as one of the hot topics in molecular biology. On the one hand, the AS isoform affected the biogenesis of miRNA and lncRNA. For example, there is a kind of tissue-specific AS regulating the biogenesis of miR-412 involved in cell death [40]. MiR846 and miR842 are regulated by abscisic acid, originating from different AS isoforms [41]. On the other hand, miRNA targeted partial AS isoforms, decreasing the regulation by miRNA [42].

In this study, the structural and transporter-coding genes related to the biosynthetic pathway of FA and cellulose were mined in *C. sativa* for the first time. Then, AS events, miRNAs and lncRNAs were analyzed, and the potential regulation in the biosynthesis of cannabinoids, FAs and cellulose was predicted. Moreover, the crosstalk between AS and miRNAs or lncRNAs was revealed. Taken together, all these results will offer candidate genes related to the biosynthetic pathway of cannabinoids, FA and cellulose and enhance the knowledge of gene expression and regulation in hemp.

## 2. Results

### 2.1. RNA-Seq Library Construction and Analysis

‘Yunma No.1’ is the first industrial hemp cultivar bred by the Yunnan Academy of Agricultural Sciences. Its THCA content is <0.3%, complying with the internationally adopted EU standards for hemp [43]. To obtain transcriptomic information of ‘Yunma No.1’, a strand-specific RNA-Seq library was constructed and sequenced by 90 bp paired-end (PE) sequencing. After filtering the low-quality reads, in total, ~66 million clean reads were obtained. More than 83% of them could be mapped to the published *C. sativa* ‘Purple Kush’ genome using tophat with the default parameter [44]. Then, the 735,445,652 mapped reads were used for assembly by Cufflinks software [45]. To predict the gene functions, the 67,035 assembly transcripts were annotated by NCBI nonredundant protein sequences (NR) and Kyoto Encyclopedia of Genes and Genomes (KEGG) databases [46,47].

### 2.2. Structural and Transporter Genes Related to the Biosynthetic Pathway of Cannabinoids, FA and Cellulose

The genes related to cannabinoids, FA and cellulose metabolism were focused on because the main products of *C. sativa* are derived from cannabinoids, FA and cellulose [3,4,5].

Based on the NR annotation, a total of 154 transcripts involved in the biosynthesis of cannabinoids, via the hexanoate, MEP and GPP pathways, were obtained. In the hexanoate pathway, the Delta12-oleic acid desaturase gene was identified [48], which contains one transcript; the Lipoxygenase (LOX) gene contains eight transcripts, and the Acyl-activating enzyme (AAE) gene contains 87 transcripts. In MEP pathway, 1-deoxy-D-xylulose-5-phosphate synthase (DXS), 1-deoxy-D-xylulose-5-phosphate reductoisomerase (DXR), 2-C-methyl-D-erythritol 4-phosphate cytidylyltransferase (MCT), 4-diphosphocytidyl-2-C-methyl-D-erythritol kinase (CMK), 2-C-methyl-D-erythritol 2,4-cyclodiphosphate synthase (MDS), 4-hydroxy-3-methylbut-2-en-1-yl diphosphate synthase (HDS) and 4-hydroxy-3-methylbut-2-enyl diphosphate reductase (HDR) genes were identified, which contain 6, 13, 7, 2, 3, 15 and 3 transcripts, respectively. In the GPP pathway, the geranyl diphosphate synthase (GPPS) gene was identified, which contains one transcript. In the cannabinoid biosynthetic pathway, the AAE gene catalyzed the hexanoate to form hexanoyl-CoA, which provided the olivetol synthase (OLS) forming olivetolic acid with substrate. The OLS gene was identified, containing one transcript. The MEP pathway supplies substrates for GPPS, and the prenyl side-chain originates from the prenyltransferase (PT) gene to form cannabigerolic acid (CBGA) [49]. GPPS and PT genes contained one and eight transcripts, respectively. Finally, THCA and CBDA are separately synthesized through THCA synthase (THCAS) and CBDA synthase (CBDAS) (Appendix A, Figure 1). 

In the FA biosynthetic pathway, firstly, the pyruvate oxidative was decarboxylated by the pyruvate dehydrogenase complex (PDHC) to form acetyl-CoA in the plastid, which was reversely catalyzed by acetyl-CoA carboxylase (ACCase) to produce malonyl-CoA [19]. In this study, 18 transcripts of PAHC and eight transcripts of ACCase were identified. For the malonyl-CoA, ACP transacylase (MCAAT) transferred malonyl-CoA to malonyl-acyl carrier protein (ACP), which gradually converted into saturated FAs by β-ketoacyl-ACP synthase (KAS), β-ketoacyl-ACP reductase (KAR), hydroxyacyl-ACP dehydrase (HAD) and enoyl-ACP reductase (EAR). The KAR gene was identified as containing one transcript. The butyryl-ACP and malonyl-ACP were catalyzed by KAS I and KAS II to form 16-carbon palmitoyl-ACP (C16:0-ACP) and 18-carbon stearoyl-ACP (C18:0-ACP). C18:0-ACP was efficiently catalyzed by stearoyl-ACP desaturase (SAD) to produce C18:1-ACP. Next, oleate desaturase (chloroplast-type) (FAD6) catalyzed C18:1-ACP to form C18:2-ACP, which was finally catalyzed by linoleate desaturase (chloroplast-type) (FAD7/8) to form C18:3-ACP [50,51]. Both FAD6 and FAD7/8 genes contained one transcript. Then, C16:0-ACP, C18:0-ACP, C18:1-ACP, C18:2-ACP and C18:3-ACP could be further catalyzed by acyl-ACP thioesterase A (FATA), acyl-ACP thioesterase B (FATB) and palmitoyl-CoA hydrolase (PCH), respectively [52], to produce free FAs (including C16:0, C18:0, C18:1, C18:2 and C18:3) [20,53]. Three unigenes encoding the FATB enzyme were identified (Appendix A, Figure 2). Among these free FAs, oleic acid (ω-9 C18:1) and linoleic acid (ω-6 C18:2) as precursor substances are involved in the hexanoate pathway of the cannabinoid synthesis pathway to synthesize the hexanoate in hemp [48,54,55,56].

In addition, 331 transcripts potentially involved in the cellulose metabolism were also identified. Cellulose synthase (CESA) can synthesize cellulose at the plasma membrane in land plants [57], which contain 27 transcripts. Sucrose synthase (SuSy) catalyzes UDP into UDP-glucose for each CESA can synthesize a single glucan chain [58], which contains 18 transcripts. The UDP-galactose and UDP-glucose transporter are reported that make substrates available for polysaccharide biosynthesis and have an important impact on maintaining cell wall integrity [59,60]. There were seven and 28 transcripts classified into UDP-galactose and UDP-glucose transporters, respectively. In the process of secretory vesicles targeting the SCW domain, the motor proteins kinesin-4 and kinesin-like proteins are involved in cell transport in microtubules [61,62], which contained four and 175 transcripts. There was a number of non-CESA proteins also involved in cellulose production, including COBRA-like protein (COBL), endo-1,4-beta-glucanase, chitinase-like protein (CTL), alpha-glucosidase I, VILLIN, katanin-like protein, glycosyltransferase-like protein KOBITO1 and inositol polyphosphate 5-phosphatase (FRA) [63,64,65,66,67,68,69,70,71,72], which contained eight, one, four, one, two, three, one and three transcripts, respectively (Appendix A).

### 2.3. Genome-Wide Identification of AS

AS events were identified using ASTALAVISTA with default options [73]. Results showed that 27,907 AS events were predicted, including 9773 intron retention (IntronR), 3404 exon skipping (ExonS), 15,970 alternative 3′ last exon (AltA) and 6388 alternative 5′ first exon (AltD) (Appendix A) [74]. These results were consistent with the four AS types that account for the majority in plants [75]. To validate the authenticity of the predicted AS, five AS events were verified by RT-PCR. The electrophoresis results were in concordance with the RNA-Seq data (Appendix A).

Then, the AS events that occurred in structural genes related to cannabinoids, FA and cellulose were analyzed. Results showed that AS occurred in the cannabinoid biosynthetic structural genes *LOX*, *AAE*, *PT*, *DXR*, *MCT* and *HDR* (Appendix A). AS also occurred in the FA biosynthetic structural genes, including *PDHC* and *ACCase* occurred AS (Appendix A) as well as the cellulose biosynthetic structural genes, including *CESA*, *CSL*, UDP-galactose transporter, UDP-glucose transporter, kinesin-like protein, kinesin-4, COBRA-like protein, *CTL*, katanin-like protein, *FRA*, *SuSy* and *MAPKKK* (Appendix A).

### 2.4. Identification of Regulatory Proteins in C. sativa

Regulatory proteins play a key role in various biological processes, such as vegetation growth, stress response and the synthesis of bioactive compounds [76,77]. To predict the regulatory proteins in *C. sativa*, the 48,799 assembly transcripts were compared with the sequences in the iTAK database. In total, 2012 tfs were identified including 63 families in which myb- > myb-related (152), bhlh (128), myb- > myb (118), c3h (114) and garp- > garp-g2-like (99) were the top five families. A total of 900 trs were found including 24 families, in which set (152), snf2 (128), others (127), phd (79) and gnat (74) were the top five families. A total of 2161 pks were identified, which included 114 families. Among them, the largest family is rlk-pelle_dlsv with 354 members (Appendix A). Among them, some regulatory proteins were potentially involved in the cellulose metabolism. For example, the mitogen-activated protein kinase kinase (MAPKKK) could promote cell division, differentiation and the synthesis of cellulose in *Arabidopsis* [78], which contains 48 transcripts. Myb domain protein 46 (myb46) enhanced the content of cellulose by targeting several secondary wall nac regulators and *cesas* in *Arabidopsis* [79], which contained one transcript (Appendix A).

### 2.5. Identification of miRNAs in C. sativa 

The small RNA library was constructed using pooled RNA of stems, roots and leaves of *C. sativa* and sequenced by the HiSeq 2000 platform. After filtering, a total of 22,164,487 clean reads ranging from 18 to 30 nucleotide (nt) in size were obtained. Among them, the clean reads with counts ≥5 were further analyzed by the psRobot and Mireap software to predict the miRNAs in *C. sativa.* After checking manually, a total of 563 miRNAs were predicted. Among them, 183 miRNAs were identified as conserved miRNAs by comparing with miRbase, which could be divided into 36 families in which miR156 was the largest family, with 16 members. The remaining 380 miRNAs were adjudged to be *C. sativa*-specific miRNAs, which were grouped into 60 families. Among them, CsmiRNA-n4 was the largest family that possessed 34 members (Appendix A).

### 2.6. Target Prediction of miRNAs in C. sativa

The psRNATarget software was used to predict the target genes of the miRNAs in *C. sativa.* As a result, the 178 conserved and 365 *C. sativa*-specific miRNAs were found to potentially target 1822 and 3951 transcripts, respectively (Appendix A). These transcripts were involved in multiple functions. Among these target transcripts, 2673 of them were annotated by the KEGG database. KEGG enrichment showed that 15, 25, 160, 271 and 29 transcripts were involved in cellular processes, environmental information processing, genetic information processing, metabolism and organismal systems, respectively, in which the top five enrichment pathways were RecQ-mediated genome instability protein 1, CCR4-NOT transcription complex subunit 4, KAS II, GPI ethanolamine phosphate transferase 3 subunit O and alpha-mannosidase (Figure 3).

Then, we considered the target genes potentially involved in the biosynthesis of cannabinoids, FA and cellulose. As a result, a total of 21 miRNAs including four conserved families of miR156/miR477/miR5658 and 17 *C. sativa*-specific miRNAs including families of CsmiRNA-n28/n33/n34/n39/n57 were found to potentially target *AAE, DXR, DXS, HDR* and *LOX* structural genes in cannabinoid biosynthesis. MiR530 and CsmiRNA-n52 f targeted *ACCase* structural genes in FA biosynthesis. Nine conserved miRNAs families including miR1508, miR171, miR172, miR482 and miR5569 and 35 *C. sativa*-specific miRNAs including families of CsmiRNA-n1/n4/n15/n22/n23/n24/n25/n27/n32/n34/n37/n56 were found to potentially target *CESA**,* COBRA-like protein, *CSL*, *FRA,* kinesin-like protein, UDP-galactose transporter and *VILLIN* genes in the biosynthesis of cellulose (Table 1). The regulatory protein genes targeted by miRNAs were also investigated. Results showed that 218, 99 and 186 miRNAs potentially targeted 337 TFs, 148 TRs and 290 PKs, respectively (Appendix A). 

Subsequently, the crosstalk between AS and miRNAs involved in cannabinoids, FA and cellulose was analyzed. As shown in Appendix A, there were two types for the miRNAs acting on the isoforms of the structural gene. One was miRNAs targeting all the isoforms of one structural gene. For example, CsmiRNA-n33j.1-3p targeted the transcripts (T_00033798/00033799/00033800) of AAE. The other was the miRNAs targeting part of isoforms of one structural gene. For example, there were six transcripts (T_00069671/00069672/00069673/00069674/00069675/00069676) of *AAE*, in which only the transcripts (T_00069673/00069674/00069676) were targeted by CsmiRNA-n33j.1-3p.

### 2.7. Cleavage Targets Guided by miRNAs

miRNAs could act on their targets by cleavage [80]. In order to validate the authentic cleavage targets of miRNAs, a degradome library was sequenced, and psRobot software was used to analyze the cleavage site of target genes and the reads of degradation fragments [81]. As a result, a total of 10,810,702 clean reads represented by 5,623,324 unique sequences were obtained after high-quality filtering and removing the adapters and null inserts. Most of the miRNAs in plants target genes at the 9th, 10th and 11th nucleotides of the complementary region between miRNA and mRNA, occasionally at the other site [82]. The cleavage sites of miRNA targets were predicted by psRobot software, as shown in Appendix A.

In total, 145 transcripts were identified, which were targeted by 62 miRNAs, including 40 conserved miRNAs and 22 *C. sativa*-specific miRNAs (Appendix A). The major targets of conserved miRNAs were TFs involved in development, responding to abiotic stresses and metabolism. In aspects of development, for example, miR156 targeted 34 squamosa promoter binding protein like transcripts (*SPLs*). In *Arabidopsis thaliana* L. *SPLs* played an important part in flowering development [83]. miR160 targeted six auxin response factor 16 (*ARF16*) transcripts. It was reported that miR160 regulated auxin mediated development by ARF10/16/17 in tomato through posttranscriptional regulation [84]. MiR164 targeted 10 CUP SHAPED COTYLEDON 2 (*CUC2*) transcripts. MiR164 regulated the axillary meristem formation by targeting *CUC1* and *CUC2*, participating in the development of cotyledons and floral organs in *Arabidopsis* [85]. In response to abiotic stresses, 20 transcripts encoding NAC TFs were identified as targets of miR164. NAC TFs were validated for response to salinity as well as high temperature and drought stresses in *Brassica juncea* L. Czern [86]. In aspects of metabolism, miR393 targeted two transport inhibitor response 1 (*TIR1*) genes and two auxin signaling f-box 2 (*AFB2*) genes. In *Arabidopsis*, *TIR1* and *AFB2* had an effect on the auxin sensitivity [87]. In addition, 16 ubiquitin-conjugating E2 enzyme transcripts were targeted by miR399. A previous study showed that miR399 targeted ubiquitin-conjugating E2 enzyme genes functioning in the control of inorganic phosphate homeostasis in *Arabidopsis* [88].

A few *C. sativa*-specific miRNAs were also found, including CsmiRNA-n56/n10/n27/n30 and other proteins. For catalysis, CsmiRNA-n56 targeted two cytokinin hydroxylase (*CYP735A1*) genes. CYP735A1 is a cytochrome P450 monooxygenase, and it was demonstrated that it catalyzed the biosynthesis of *trans*-Zeatin in *Arabidopsis* [89]. CsmiRNA-n10 targeted the 3-hydroxy-3-methylglutaryl-coenzyme A reductase gene, which catalyzed the HMG-CoA to form MVA [90]. CsmiRNA-n27 targeted the U-box domain-containing protein, which participated in the initial fruit developmental stage in *Musa acuminate* L [91]. In addition, CsmiRNA-n30 targeted tubby-like F-box protein, which functioned in abiotic stress tolerance in *Manihot esculenta* Crantz [92]. Furthermore, CsmiRNA-n30 also targeted *SURP* and G-patch domain-containing protein 1-like protein (*SUGP1*). SURP and SUGP1 were associated with plasma LDL cholesterol levels, coronary artery disease and some other energy metabolism [93].

### 2.8. Identification of lncRNAs in C. sativa

To decipher the functions of lncRNAs in *C. sativa*, a pipeline was firstly established to predict the lncRNAs. After removing 48,800 transcripts without Nr annotations, the 1375 transcripts less than 300 bp in length were discarded. Then, the remaining transcripts were filtered by ORF ≤ 100 aa using ESTScan, leaving 8260 genes. Next, the CPAT was used to filter the 430 possible protein-coding genes, leaving a total of 7830 transcripts. Finally, BLASTN was used to remove the housekeeping npcRNAs (47) from the alignment with the Rfam database, thereby obtaining 7783 lncRNA candidates and 47 housekeeping npcRNAs (Appendix A, Figure 4A).

After that, an assay of the length distribution of lncRNAs was conducted. It was found that 5834 lncRNAs ranged from 300 to 1000 bp. Most of the lncRNAs were distributed between 1000 and 2000 bp. The others included 525 lncRNA candidates with 2000–4000 bp and 41 over 4000 bp in length (Figure 4B).

### 2.9. Target Prediction for lncRNAs in C. sativa

BLASTN was used to predict the targets of lncRNAs to reveal their regulatory role in *C. sativa*. Among the 7783 lncRNAs, 1516 lncRNAs were revealed to target 3452 transcripts (Appendix A). Among the 3452 transcripts, 1536 were annotated by the KEGG database. KEGG enrichment showed that 26, 25, 102, 224 and 25 genes were involved in cellular processes, environmental information processing, genetic information processing, metabolism and organismal systems, respectively, in which the top five enrichment pathways were 1,4-alpha-glucan branching enzyme, uroporphyrinogen-III synthase, RecQ-mediated genome instability protein 1, structure-specific endonuclease subunit SLX1 and flap endonuclease-1 (Appendix A).

Then, the targets related to the biosynthetic pathway of cannabinoids, FA and cellulose were focused on. Therefore, lncTar software was used to further predict the interactions between lncRNAs and target genes. As a result, nine lncRNAs were found to target the *LOX, GPPS, AAE* and *DXS* genes in the cannabinoid biosynthetic pathway, and eight lncRNAs were found to target the *COBL*, kinesin-like and *MAPKKK* genes in the cellulose metabolism (Table 2). In addition, 111 TFs, 59 TRs and 141 PKs were also identified, which were targeted by 92, 90 and 135 lncRNAs, respectively (Appendix A).

Subsequently, the crosstalk between AS and lncRNAs related to the cannabinoids and cellulose were analyzed. For lncRNAs, on the one hand, all of the isoforms (T_00013932 and T_00013933) targeted all the kinesin-like protein gene isoforms (T_00013935/13936/13937). On the other hand, part isoforms of lncRNAs (T_00056848/56849/56852/56853/56854 and T_00020067) targeted one transcript of GPPS (T_00049365) and DXS (T_00064406). For target genes, all of the *LOX* isoforms (T_00022090 and T_00022091) were targeted by different lncRNAs (T_00010712 and T_00019445), and parts of AS of kinesin-like protein gene isoforms (T_00043674/43675/43677/43678/43681) were targeted by different lncRNAs (T_00043682/29565/77089/43682), respectively (Appendix A).

### 2.10. QRT-PCR Analysis of the Expression Files of lncRNAs and Target Genes

Seven lncRNA/structural gene pairs were selected for qRT-PCR validation, and the primer sequences are shown in Appendix A. As a result, three lncRNA/structural gene pairs were identified with negative expression files in at least four tissues, including lncR880 (T_00090880)/*AAE* (T_00058852), lncR682 (T_00043682)/kinesin-like protein (T_00043674) and lncR578 (T_00030548)/*COBL* (T_00016031) (Figure 5), indicating the negative regulation of these lncRNAs.

## 3. Discussion

In this study, transcriptome, small RNA and degradome libraries of hemp ‘Yunma No.1’ were constructed and comprehensively analyzed. Compared with previous studies in *C. sativa*, three advancements were made. Firstly, the protein-coding genes related to cannabinoids, FA and cellulose were mined, and the transporter and regulatory protein genes were systematically predicted. 

Secondly, to reveal the gene expression and regulation at post-transcriptional level, the AS, miRNAs and lncRNAs in *C. sativa* firstly were predicted. A total of 27,907 AS events in *C. sativa* were identified, and five AS events were validated by RT-PCR at random. Among them, AS occurred in some structural genes of the cannabinoids, FA and cellulose biosynthetic pathway. 

MiRNAs and lncRNAs functioned in multiple pathways. We focused on the top five KEGG enrichment pathways of miRNA/lncRNA targets. For miRNAs, it was reported that miR-362 targeted RecQ-mediated genome instability protein 1 [94]. miR-125b targeted ER alpha-1, 2-mannosidase and played a critical role in maintaining protein homeostasis in the mammalian secretory pathway [95], while, KAS II, GPI ethanolamine phosphate transferase 3 subunit O and CCR4-NOT transcription complex subunit 4 potentially targeted by miRNAs were not reported. KAS II was involved in fatty acid elongation from 16:0-ACP to 18:0-ACP [96]. GPI ethanolamine phosphate transferase 3 subunit O functioned in GPI-anchor biosynthesis [97]. The top five KEGG enrichment pathways targeting lncRNAs have not been revealed to date. For example, 1,4-alpha-glucan branching enzyme played a significant role in the biosynthesis of branched polysaccharides, glycogen and amylopectin [98]. All these results indicate the important regulation of miRNAs and lncRNAs in *C. sativa*. 

MiRNAs and lncRNAs potentially regulate the biosynthesis of cannabinoids, FA and cellulose in two aspects. On the one hand, they directly target the structural genes (Table 1 and Table 2, Figure 5). On the other hand, they targeted the structural gene related regulatory protein genes. Previously, we determined some GA- or isoflavone-related TFs, TRs and PKs potentially targeted by miRNAs, indicting the significant roles of regulatory protein genes in metabolism [77,99]. Based on the known biosynthetic pathways of cannabinoids, Luo et al. successfully biosynthesized cannabinoids along with the unnatural analogues in yeast [100], but the different isoforms of structural genes, regulatory protein genes and non-coding RNAs were not considered. In this study, only a strand-specific RNA-Seq library was constructed. In the future, more libraries should be sequenced, and co-expression analysis will be carried out between the interested structural genes and regulatory genes to screen out and validate highly relevant regulators and make full use of them. 

Additionally, the crosstalk between AS and miRNAs or lncRNAs in the biosynthesis of cannabinoids, FA and cellulose was studied for the first time in *C. sativa* (Appendix A). AS could regulate miRNA and lncRNA-mediated gene regulation by producing mRNA isoforms, which contained miRNAs/non-coding RNAs target sites or not [101]. A previous study revealed that the 44 of 354 identified miRNA binding sites were affected by AS in *Arabidopsis* [42]. Some lncRNA genes also possessed introns, leading to the production of lncRNA isoforms [102]. In *Arabidopsis*, Calixto et al. identified 135 cold-responsive lncRNAs, of which AS occurred in one-third [103]. In this research, the crosstalk between 17 lncRNAs and their target transcripts was predicted in *C. sativa* (Appendix A). 

During our work, a PacBio SMRT and a nanopore-based assembly genome of *C. sativa* was released in the NCBI under the accession numbers PRJNA73819 and GCA_900626175.2, providing good sources for transcriptome assembly of *C. sativa* for the future [16,17].

Taken together, all these results enhance the knowledge of gene expression and regulation in *C. sativa*.

## 4. Materials and Methods

### 4.1. Plant Materials and RNA Extraction

Hemp ‘Yunma No.1’ plants were cultivated in field conditions at the Yuanjiang experimental station of Bast Fiber Crops, Chinese Academy of Agricultural Sciences. Different tissues including flowers, fruits, stems, leaves and roots were collected from three female hemp plants in September 2014 and quickly frozen in liquid nitrogen. Total RNAs of fruits and roots were extracted using Trizol Reagent (Invitrogen, Carisbad, CA, USA). Total RNAs of flowers, stems and leave were extracted using Lysis Buffer PL (Bioteke, Wuxi, China). Then, these RNAs were treated by RQ1 DNase (Promega, Madison, WI, USA) to eliminate the genome DNA. Briefly, about 100 μg total RNA of each sample was incubated with 20 U DNase I at 37 °C for 30 min. Then, equal volumes of phenol/chloroform/isoamyl alcohol (25:24:1) were added and mixed. After centrifugation at 12,000 rpm for 5 min at room temperature, the upper layer was transferred to a new tube. Subsequently, 1/10 volume of 3 M sodium acetate and 2 volumes of chilled ethanol were added, mixed and kept at for 20 min at −80 °C. Finally, after centrifugation at 12,000 rpm for 10 min at 4 °C, the precipitation was washed with chilled 70% ethanol, dried and dissolved in a suitable amount of DEPC-treated water. Equal quantities of total RNAs from different tissues were pooled together, then their integrity and concentrations were examined using a NanoDrop 2000c bioanalyzer (Thermo Fisher Scientific, Waltham, MA, USA). The pooled RNA with an integrity number of more than seven was used for the following RNA-Seq, small RNA and degradome library construction. 

### 4.2. RNA-Seq Library Construction and Analysis

The RNA-Seq library was built by using the dUTP method according to the manufacturer’s protocol [104]. First, mRNA was isolated from 2 μg total RNA by using Oligo (dT) magnetic beads. Then, about 20 ng of mRNA was used for library construction. After the RNA sample was qualified, the mRNA was enriched with magnetic beads with Oligo (dT). Fragmentation buffer was used for breaking mRNA into short fragments, which were used as a template to synthesize one-strand cDNA with six base random primers (random hexamers). Then, we added buffer, dNTPs (the dTTP in dNTP was replaced by dUTP) and DNA Polymerase I and RNase H to synthesize two-strand cDNA. The double-strand cDNA was purified by AMPure XP beads. The second strand of cDNAs containing Us was degraded by USER enzyme and further repaired, A-tailed and connected to the sequencing adapter. Finally, PCR amplification was performed, and the final library was purified by AMPure XP beads. After the library was constructed, Qubit was used for preliminary quantification, and then Agilent 2100 was used to detect the size of the insert in the library. High-throughput sequencing was carried out by using a HiSeq 2000 platform. Raw reads were first processed using Trimmomatic with the following options: ILLUMINACLIP: TruSeq3-PE. fa: 2:30:10 LEADING:5 TRAILING:5 to remove adapters and discard reads containing low quality or N bases (below quality 5). After filtering, the clean reads were aligned with the *C. sativa* ‘Purple Kush’ genome [18] using tophat with the -a/--min-anchor 8 option, which required that the anchor length should be more than 8bp for splicing-junction reads [18,44]. Then, the mapped reads were assembled by Cufflinks software using -F 0.05 -A 0.01 -I 15000 [45]. Subsequently, the assembled files were used to identify the AS events and types by ASTALAVISTA with default options using the following command: /usr/java/jdk1.6.0_45/jre/bin/java -Xmx4G -jar/srv/web/cgi -bin/astal avista/jar/gphase5.jar astalavista.gtf -nonmd -genome csa -output gtf -html -html 2 > &1 [73]. To determine gene functions, the assembly transcripts were annotated by NR and KEGG databases.

### 4.3. Validation of AS through Reverse Transcriptional PCR

Total RNA (2 μg) as described for RNA-Seq library construction was used for reverse transcription PCR (RT-PCR) to validate the AS events. Briefly, RT was carried out using 2 μg of total RNA by 200 U M-MLV Reverse Transcriptase (TaKaRa, Dalian, China) in a 20 μL volume under the following the conditions: 65 °C for 5 min; 25 °C for 10 min; 42 °C for 60 min and 70 °C for 15 min. The resulting cDNA was used for PCR amplification, which was carried out under the following conditions: 95 °C for 3 min; 45 cycles of 94 °C for 30 s; 58 °C for 30 s and 72 °C for 15 s. The PCR products were separated by electrophoresis with a 3% agarose gel. The primers are listed in Appendix A.

### 4.4. Small RNA Library Construction and Analysis 

The small RNA library was constructed as described [105]. Briefly, 10–30-nt small RNAs were purified from 1 μg Total RNA by a 15% denaturing polyacrylamide gel and then ligated to the adapters. After reverse transcription by M-MLV (TaKaRa, Dalian, China), these small RNAs were amplified by PCR and isolated by 6% polyacrylamide gel, and suitable sizes were recovered and checked by qPCR. Finally, the library was sequenced by the HiSeq 2000 platform at Beijing HiTseq Technology Co. Ltd. (Beijing, China).

After sequencing, the raw reads were first processed using Trimmomatic software [106] with the option LEADING:5 TRAILING:5 to discard reads containing low quality or N bases (below quality 5). Then, adaptor and shorter reads (less than 16 nt) were removed by fastx_clipper in the FASTX-Toolkit v0.0.13 (http://hannonlab.cshl.edu/fastx_toolkit/) (2 February 2010) using the option fastx_clipper -Q 33 -a CTGTAGGCACCATCAATCA -l 16 -d 0 -n -v -M 4. Then, the clean small RNAs with counts ≥5 ranging from 18 to 30 nt were further analyzed. These small RNAs were compared with the *C. sativa* ‘Purple Kush’ genome sequences by the psRobot software (v1.2) [81] and mireap software [107] using the default parameters. Then, these predicted miRNAs were checked manually according to the criterion of miRNAs [108]. Minimum free energy index (MFEI) was also calculated [109], and a cutoff >0.85 was used to distinguish miRNAs from protein-coding genes because the MFEI of more than 90% of miRNA precursors is >0.85.

### 4.5. Prediction of MiRNA Targets

Through the psRNATarget Server using an expectation penalty score of ≤3, targets of miRNAs were predicted [110].

### 4.6. Degradome Library Construction, Sequencing and Analysis

Briefly, about 150 μg pooled total RNA was used for degradome library construction by BGI (Shenzhen, China) following a published protocol [111]. In brief, mRNAs were isolated using oligo(dT) magnetic beads. Then, single-stranded 5’ RNA adaptors were ligated to mRNA fragments using T4 RNA ligase (Ambion, Austin, TX, USA) and then reverse-transcribed into cDNA using Superscript III reverse transcriptase (Invitrogen, Carlsbad, CA, USA). After digestion with MmeI, 3’ DNA adaptors were ligated to the digested DNA fragments. Finally, the products were amplified by PCR and sequenced using the HiSeq 2000 system (BGI, Shenzhen, China). The cleavage sites and the targets of miRNAs were identified using the psRobot software (v1.2) using the options -ts 2.5 -fp 2 -tp 17 -gl 17 -p 1 -gn 1 [81]. 

### 4.7. Prediction of LncRNAs

A pipeline was established for identification of lncRNAs in hemp (Figure 4). First, the assembly transcripts without Nr annotation were further analyzed. Then, the transcripts less than 300 base pairs (bp) in length were removed. Subsequently, the remaining transcripts were filtered by ESTScan with a cutoff of ORF ≥ 100 amino acids (aa) [112]. Next, the coding potential assessment tool (CPAT) was used to filter the possible protein coding genes [113]. Finally, BLASTN program was used to remove the housekeeping npcRNAs annotated by the Rfam database.

### 4.8. Prediction of LncRNA Targets

Targets of lncRNAs were predicted by using BLASTN to search the lncRNAs with regions significantly complementary to the protein-coding genes using the cutoff 1 × 10^−5^ [114,115]. Then, the interesting lncRNA-RNA interactions were further analyzed by LncTar using the cutoff of the normalized deltaG (ndG) ≤ 0.1 [116].

### 4.9. QRT-PCR Analysis of the Expression Files of LncRNAs and Targets

RNA purity, concentration and integrity of each sample treated by DNase I were examined using a NanoDrop 2000c bioanalyzer (Thermo Fisher Scientific, Waltham, MA, USA). Then, RT was carried out using 1 μg total RNA and 200 U M-MLV Transcriptase (TaKaRa, Dalian, China) in a 20 μL volume. The reaction was conducted at 70 °C for 10 min, 42 °C for 60 min and 70 °C for 15 min. The RT product was diluted 40 times with sterile water. The qPCR was conducted on the BIO-RAD CFX system in a 20 μL volume containing 4 μL diluted cDNA, 250 nM forward primer, 250 nM reverse primer, and 1 × SYBR Premix Ex Taq II (TaKaRa, Dalian, China) using the following conditions: 95 °C for 3 min, 40 cycles of 95 °C for 15 s, 59 °C for 15 s and 72 °C for 15 s. Each test was performed in triplicate. The 40S ribosomal protein S8 (T_00004056) was used as an internal reference because its expression level is relatively stable in different tissues. Melting curves were analyzed to verify the specificity using the Bio-Rad CFX Manager software. The relative expression levels were calculated using the 2^−ΔΔCT^ method [117]. The primer sequences are listed in Appendix A.

## Figures and Tables

**Figure 1 ijms-22-11989-f001:**
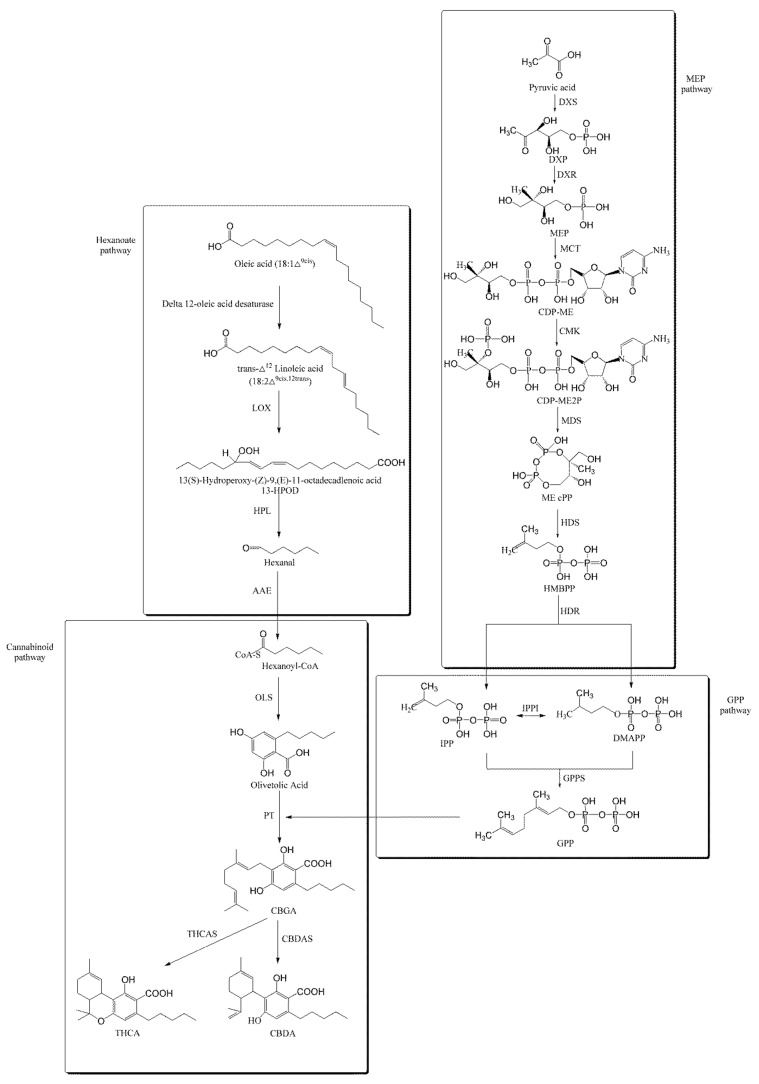
Proposed THCA and CBDA biosynthetic pathway in *C. sativ**a*. LOX: lipoxygenase; HPL: hydroperoxide lyase; AAE: acyl-activating enzyme; DXS: 1-deoxy-D-xylulose-5-phosphate synthase; DXR: 1-deoxy-D-xylulose 5-phosphate reductoisomerase; MCT: 2-C-methyl-D-erythritol 4-phosphate cytidylyltransferase; CMK: 4-diphosphocytidyl-2-C-methyl-D-erythritol kinase; MDS: 2-C-methyl-D-erythritol 2,4-cyclodiphosphate synthase; HDS: 4-hydroxy-3-methylbut-2-en-1-yl diphosphate synthase; HDR: 4-hydroxy-3-methylbut-2-enyl diphosphate reductase; IPPI: Isopentenyl diphosphate isomerase; GPPS: Geranyl diphosphate synthase; OLS: olivetol synthase; PT: prenyltransferase; THCAS: tetrahydrocannabinolic acid synthase; CBDAS: cannabidiolic acid synthase.

**Figure 2 ijms-22-11989-f002:**
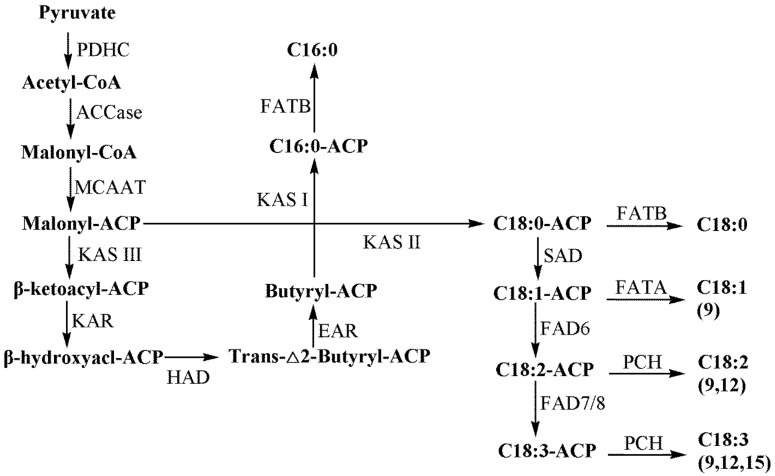
Proposed fatty acid biosynthetic pathway in *C. sativa* [20]. PDHC: pyruvate dehydrogenase complex; ACCase: acetyl-CoA carboxylase; MCAAT: malonyl-CoA: ACP transacylase; KAS: β-ketoacyl-ACP synthase; KAR: β-ketoacyl-ACP reductase; HAD: hydroxyacyl-ACP dehydrase; EAR: enoyl-ACP reductase; SAD: stearoyl-ACP desaturase; FAD6: oleate desaturase (chloroplast-type); FAD7/8: linoleate desaturase (chloroplast-type); FATA/B: acyl-ACP thioesterase A/B; PCH: palmitoyl-CoA hydrolase.

**Figure 3 ijms-22-11989-f003:**
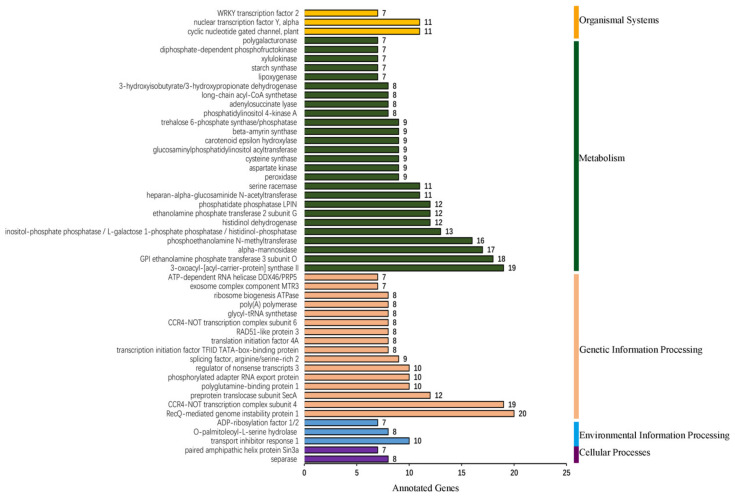
KEGG enrichment of miRNA targets.

**Figure 4 ijms-22-11989-f004:**
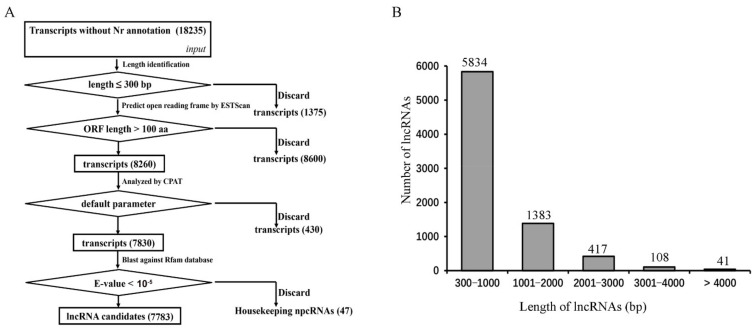
The schematic pipeline and size distribution of lncRNAs. (**A**) Predicted pipeline of lncRNA candidates. (**B**) Size distribution of lncRNA candidates.

**Figure 5 ijms-22-11989-f005:**
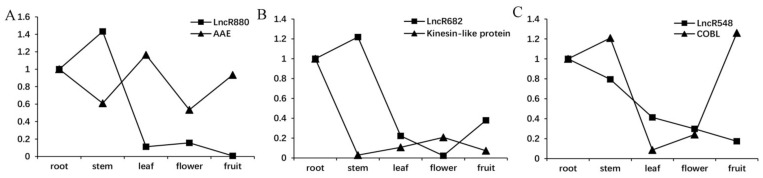
QRT-PCR validation of expression profiles of lncRNA/structural gene pairs of lncR880/AAE (**A**), lncR682/kinesin-like protein (**B**) and lncR578/COBL (**C**). The average expression profile of every gene in the root was chosen as a control and set to 1.

**Table 1 ijms-22-11989-t001:** MiRNAs target structural genes potentially involved in cannabinoids, fatty acid and cellulose biosynthesis.

miRNA Name	Target ID	Target Annotation	Expectation	Pathway
CsmiRNA-n33j.1-3p	T_00033798	AAE	3	cannabinoids
	T_00033799		3	
	T_00033800		3	
	T_00007721		3	
	T_00007722		3	
	T_00069673		3	
	T_00069674		3	
	T_00069676		3	
miR5658	T_00030776		3	
	T_00030777		3	
	T_00030778		3	
	T_00030779		3	
miR477a-5p	T_00014806		3	
miR477b-5p			3	
miR477a-5p	T_00014807		3	
miR477b-5p			3	
miR156e-3p	T_00050643		1.5	
CsmiRNA-n39d.1-3p	T_00043759	DXR	3	
CsmiRNA-n39e.1			3	
CsmiRNA-n34g.2	T_00038493	DXS	2.5	
CsmiRNA-n34a-5p	T_00064406		0	
CsmiRNA-n34b-5p			0.5	
CsmiRNA-n34c-5p			1	
CsmiRNA-n34d.1-5p			2	
CsmiRNA-n34d.2			3	
CsmiRNA-n34e.2-3p			0.5	
CsmiRNA-n34e.2-5p			0.5	
CsmiRNA-n34f.1-5p			0.5	
CsmiRNA-n34f.2-5p			3	
CsmiRNA-n34h.2-3p			1.5	
CsmiRNA-n57-5p			2.5	
CsmiRNA-n28	T_00018927	HDR	1.5	
	T_00018928		1.5	
	T_00018929		1.5	
miR156e-3p	T_00090667	LOX	3	
CsmiRNA-n33e-3p	T_00023464		2.5	
	T_00023465		2.5	
	T_00023466		2.5	
CsmiRNA-n52a	T_00067318	ACCase	3	fatty acid
CsmiRNA-n52d			3	
CsmiRNA-n52e			3	
CsmiRNA-n52f			3	
CsmiRNA-n52a	T_00067319		3	
CsmiRNA-n52d			3	
CsmiRNA-n52e			3	
CsmiRNA-n52f			3	
CsmiRNA-n52a	T_00067320		3	
CsmiRNA-n52d			3	
CsmiRNA-n52e			3	
CsmiRNA-n52f			3	
miR530-3P	T_00087904		3	
CsmiRNA-n27a.2	T_00016027	COBL	1	cellulose
CsmiRNA-n27c.2			2.5	
CsmiRNA-n27d-3p			2	
CsmiRNA-n27e-3p			2	
CsmiRNA-n27f			1.5	
CsmiRNA-n27g-3p			0.5	
CsmiRNA-n27i-3p			1.5	
CsmiRNA-n27l			0.5	
CsmiRNA-n27n-3p			0.5	
CsmiRNA-n27a.1	T_00016031		2.5	
CsmiRNA-n27b			0	
CsmiRNA-n27c.1			0.5	
CsmiRNA-n27d-5p			1	
CsmiRNA-n27e-5p			0	
CsmiRNA-n27g-5p			2.5	
CsmiRNA-n27h			3	
CsmiRNA-n27i-5p			2.5	
CsmiRNA-n27j			1.5	
CsmiRNA-n27k			1.5	
CsmiRNA-n27m			1.5	
CsmiRNA-n27n-5p			1.5	
CsmiRNA-n27o			1	
CsmiRNA-n27p			2	
CsmiRNA-n56b.1	T_00016799	Kinesin-like protein	2	
miR5569b	T_00035890		3	
CsmiRNA-n1a-5P			3	
miR5569b	T_00035891		3	
CsmiRNA-n1a-5P			3	
miR5569b	T_00035892		3	
CsmiRNA-n1a-5P			3	
miR5569b	T_00035893		3	
CsmiRNA-n1a-5P			3	
miR5569b	T_00035894		3	
CsmiRNA-n1a-5P			3	
miR5569b	T_00035895		3	
CsmiRNA-n1a-5P			3	
CsmiRNA-n22c-5p	T_00043674		3	
CsmiRNA-n22c-5p	T_00043675		3	
CsmiRNA-n22c-5p	T_00043676		3	
CsmiRNA-n22c-5p	T_00043677		3	
CsmiRNA-n25m.1	T_00051204		3	
CsmiRNA-n37a-3p	T_00068566		3	
CsmiRNA-n37b-3p			3	
CsmiRNA-n37a-3p	T_00068567		3	
CsmiRNA-n37b-3p			3	
miR5569b	T_00078801		3	
CsmiRNA-n1a-5P			3	
miR5569b	T_00078802		3	
CsmiRNA-n1a-5P			3	
miR5569b	T_00078803		3	
CsmiRNA-n1a-5P			3	
miR5569b	T_00078805		3	
CsmiRNA-n1a-5P			3	
CsmiRNA-n23d-5p	T_00080222		3	
	T_00080223		3	
	T_00080224		3	
	T_00087551		3	
	T_00087552		3	
	T_00087553		3	
	T_00087554		3	
	T_00087555		3	
	T_00087556		3	
	T_00087557		3	
	T_00087558		3	
	T_00087559		3	
	T_00087560		3	
miR482a-3p	T_00093433	CESA	3	
miR482b-3p			3	
miR482c-3p			3	
miR482a-3p	T_00093434		3	
miR482b-3p			3	
miR482c-3p			3	
miR482a-3p	T_00093435		3	
miR482b-3p			3	
miR482c-3p			3	
miR1508-5P	T_00012561		3	
CsmiRNA-n34c-3p	T_00012569		3	
CsmiRNA-n15b-5p	T_00054754	VILLIN	3	
miR172a-3p	T_00062735	FRA	3	
CsmiRNA-n4b-5p			3	
miR172a-3p	T_00062736		3	
CsmiRNA-n4b-5p			3	
CsmiRNA-n24m.1	T_00064946	CSL	2	
CsmiRNA-n32e.2	T_00019352	UDP-galactose transporter	2.5	
miR171a-5p	T_00021321		3	
miR171b-5p			3	
miR171c-5p			3	

AAE: acyl-activating enzyme; DXR: 1-deoxy-D-xylulose 5-phosphate reductoisomerase; DXS: 1-deoxy-D-xylulose-5-phosphate synthase; HDR: 4-hydroxy-3-methylbut-2-enyl diphosphate reductase; LOX: lipoxygenase; ACCase: acetyl-CoA carboxylase; COBL: COBRA-like protein; CESA: Cellulose synthase; FRA: inositol polyphosphate 5-phosphatase; CSL: Cellulose synthase-like protein.

**Table 2 ijms-22-11989-t002:** LncRNAs targeted structural genes potentially involved in cannabinoids and cellulose biosynthesis.

LncRNA ID	Target ID	Target Annotation	Start Position LncRNA	End Position LncRNA	Start Position Target	End Position Target	Pathway
T_00010712	T_00010715	LOX	1	228	1261	1488	cannabinoids
T_00010712	T_00022090		1	228	1947	2174	
T_00019445			127	556	1	430	
T_00019445	T_00022091		127	556	1	430	
T_00056848	T_00049365	GPPS	1	164	1784	1947	
T_00056849			1	164	1784	1947	
T_00056852			1	162	1786	1947	
T_00056853			1	161	1787	1947	
T_00056854			1	153	1795	1947	
T_00090880	T_00058852	AAE	1	369	19	387	
T_00020067	T_00064406	DXS	383	863	1	481	
T_00013932	T_00013935	Kinesin-like protein	1	1075	1	1075	cellulose
T_00013933			1	1066	10	1075	
T_00013932	T_00013936		1	1079	1	1079	
T_00013933			1	1070	10	1079	
T_00013932	T_00013937		1	1077	1	1077	
T_00013933			1	1068	10	1077	
T_00052635	T_00016799		558	982	1	425	
T_00043682	T_00043674		174	1620	1	1447	
T_00029565	T_00043675		357	542	1	186	
T_00077089			501	638	1	138	
T_00043682	T_00043677		174	1620	1	1447	
T_00043682	T_00043678		174	1620	1	1447	
T_00043682	T_00043681		174	1620	1	1447	
T_00030548	T_00016031	COBL	1	346	957	1302	
T_00059592	T_00061939	MAPKKK	1	1031	2317	3347	

GPPS: Geranyl diphosphate synthase; MAPKKK: mitogen-activated protein kinase kinase.

## Data Availability

All the raw data of the transcriptome and small RNAs are deposited in the NCBI short read archive (SRA) under the accession numbers SRR12904745 and SRR12886428.

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
