# Peer review of "Genome-Wide Analysis of Alternative Splicing and Non-Coding RNAs Reveal Complicated Transcriptional Regulation in Cannabis sativa L."

_ijms, 2021, doi:10.3390/ijms222111989_

Round 1

Reviewer 1 Report

The problems addressed by Authors are generally important and significant to understand, fatty acids, cellulose and cannabinoids biosynthesis in C. sativa, however the main weakness is that material and method section is not clearly enough described, that may result in problems with research reproducibility. Although results concerning targets of miRNA and lncRNA are performed mainly by bioinformatic tools, they still provide interesting starting point to guide the future research.

Sometimes Authors cite some research in materials and methods sections but not always is it possible to find there details of method or software settings. The main problems related to such category are as follows:

Paragraph 4.2

Concentration and volume of cDNA library

Cut-off values for low quality sequences.

Settings used in tophat for alignment with C. sativa genome.

Settings of ASTALAVISTA to find AS events.

Paragraph 4.3

Assessment of RNA purity, concentration and integrity

Details of DNase treatment to remove putative remnants of genomic DNA

Reverse transcription reaction; details of reaction-temperature and time, volume and amount of used RNA

qPCR protocol: conditions of PCR reaction, volume of reaction, concentration of magnesium ions, dNTPs, DNA polymerase type and concentration, additives as SYBR Green, DMSO etc. Name and manufacturer of qPCR instrument.

Justification of RT-PCR reference gene choice for example by bestKeeper software- length of reference PCR product

Software used to analyse RT-PCR results, statistical method.

Citation of method used to calculate RT-PCR results- for example Livak and Schmittgen 2001 or other.

Paragraph 4.4

Settings of Trimmomatic software to remove low quality sequences.

Why the cut-off 0.8 was used to distinguish miRNA from cDNA sequences?

Paragraph 4.6

Concentration and volume of cDNA library

Cut-off values for low quality sequences.

Settings values of psRobot software.

Paragraph 4.8

Values of LncTar used in analysis

Paragraph 4.9

The same comments as for paragraph 4.3

Other

Paragraph 2.5

Is the composed library build from the same quantity of RNA from roots, stems and leaves or the proportions between  RNA from these sources are different? It should be stated in material and methods section.

Captcha and contents of tables S19 and S20 are exchanged .

Author Response

Comments for reviewer 1:

The problems addressed by Authors are generally important and significant to understand, fatty acids, cellulose and cannabinoids biosynthesis in C. sativa, however the main weakness is that material and method section is not clearly enough described, that may result in problems with research reproducibility. Although results concerning targets of miRNA and lncRNA are performed mainly by bioinformatic tools, they still provide interesting starting point to guide the future research.

Sometimes Authors cite some research in materials and methods sections but not always is it possible to find there details of method or software settings. The main problems related to such category are as follows:

Paragraph 4.2

Concentration and volume of cDNA library

Response: Thank you for the suggestion. First, mRNA was isolated from 2 μg total RNA by using Oligo (dT) magnetic beads. Then about 20 ng mRNA was used for library construction. After the RNA sample was qualified, the mRNA is enriched with magnetic beads with Oligo (dT). fragmentation buffer was used for breaking mRNA into short fragments, which was used as template to synthesize one-strand cDNA with six base random primers (random hexamers). Then we added buffer, dNTPs (dTTP in dNTP is replaced by dUTP) and DNA Polymerase I and RNase H to synthesize two-strand cDNA. The double-strand cDNA was purified by AMPure XP beads. The second strand of cDNAs containing Us were degraded by USER enzyme and further repaired, A-tailed and connected to the sequencing adapter. Finally, PCR amplification was performed, and final library was purified by AMPure XP beads. After the library was constructed, Qubit was used for preliminary quantification, and then Agilent 2100 was used to detect the size of the insert in the library.

Cut-off values for low quality sequences.

Response: We are sorry for missing the cut-off value in the first version. Raw reads were first processed using trimmomatic with option: ILLUMINACLIP: TruSeq3-PE. fa: 2:30:10 LEADING:5 TRAILING:5 to remove adapters, and discard reads containing low quality or N bases (below quality 5).

Settings used in tophat for alignment with C. sativa genome.

Response: Thank you for the suggestion. We added the parameter “with -a/--min-anchor 8 option, which required anchor length should be more than 8bp for splicing-junction reads.”

Settings of ASTALAVISTA to find AS events.

Response: Thanks for the reminder. We added the settings as “with default option using the following command: /usr/java/jdk1.6.0_45/jre/bin/java -Xmx4G -jar /srv/web/cgi-bin/astalavista/jar/gphase5.jar astalavista.gtf -nonmd -genome csa -output gtf -html -html 2>&1.”

Paragraph 4.3

Assessment of RNA purity, concentration and integrity

Response: Thanks for the suggestion. I have revised it as described in Paragraph 4.1.

Details of DNase treatment to remove putative remnants of genomic DNA

Response: Briefly, about 100μg total RNA of each sample was incubated with 20 U DNase I at 37℃ for 30 min. Then, equal volume phenol/chloroform/isoamyl alcohol (25: 24: 1) were added and mixed. After centrifugation at 12,000 rpm for 5 min at room temperature, the the upper layer was transferred to a new tube. Subsequently, 1/10 volume of 3 M sodium acetate and 2 volume of chilled ethanol were added, mixed and kept it for 20 min at - 80℃. Finally, after centrifugation at 12,000 rpm for 10 min at 4℃, the precipitation was washed with chilled 70% ethanol, dried and dissolved in a suitable amount of DEPC-treated water.

Reverse transcription reaction; details of reaction-temperature and time, volume and amount of used RNA.

Response: We rewrote the paragraph as “Total RNA as described for RNA-Seq library construction were used for reverse transcription PCR (RT-PCR) to validate the AS events [1].

RT was carried out using 2 μg of total RNA by 200 U M-MLV Reverse Transcriptase (TaKaRa) in a 20 μl volume under the following the conditions: 65 °C for 5 min; 42°C for 60 min and 70 °C for 15 min. The resulting cDNA was used for PCR amplification, which was carried out using the following conditions: 95 °C for 3 min; 45 cycles of 94 °C for 30 s; 58 °C for 30 s; and 72 °C for 15 s. The PCR products were separated by electrophoresis with a 3% agarose gel. The primers are listed in Table S19.”

 Paragraph 4.4

Why the cut-off 0.85 was used to distinguish miRNA from cDNA sequences?

Response: because the MFEI of more than 90% miRNA precursors >0.85.

Small RNA library

Briefly,10–30-nt small RNAs were purified from 1μg Total RNA by a 15% denaturing polyacrylamide gel and then ligated to the adapters. After reverse transcription by M-MLV (TaKaRa, Dalian, China), these small RNAs were amplified by PCR and isolated by 6% polyacrylamide gel, and suitable sizes were recovered and checked by qPCR. Finally, the library was sequenced by HiSeq X Ten platform (Illumina,SanDiego,CA,USA).

Settings of Trimmomatic software to remove low quality sequences.

Response: We apologized for this missing option. We updated this part in the revised manuscript. Firstly, raw reads were first processed using trimmomatic with option: LEADING:5 TRAILING:5 to discard reads containing low quality or N bases (below quality 5). Then adaptor and shorter reads (less than 16 nt) were removed by fastx_clipper in FASTX-Toolkit (http://hannonlab.cshl.edu/fastx_toolkit/) using the option: fastx_clipper -Q 33 -a CTGTAGGCACCATCAATCA -l 16 -d 0 -n -v -M 4.

Paragraph 4.6

Settings values of psRobot software.

Response: Thanks for the suggestion. We also added the description for degradome libiary constriction “About 150 μg pooled total RNA was used for degradome libiary constriction by BGI (Shenzhen, China) following a published protocol [2]. In brief, mRNAs were isolated using oligo(dT) magnetic beads. Then, single‐stranded 5' RNA adaptors were ligated to mRNA fragments using T4 RNA ligase (Ambion, Austin, TX, USA) and then reverse‐transcribed into cDNA using Superscript III reverse transcriptase (Invitrogen, Carlsbad, CA, USA). After digestion with MmeI, 3' DNA adaptors were ligated to the digested DNA fragments. Finally, the products were then amplified by PCR and sequenced using HiSeq 2000 system (BGI, Shenzhen, China). ”

The option of psRobot software is -ts 2.5 -fp 2 -tp 17 -gl 17 -p 1 -gn 1.”

Paragraph 4.8

Values of LncTar used in analysis

 Response: we added “using the cutoff of the normalized deltaG (ndG) ≤ 0.1.”

Paragraph 4.9

qPCR protocol: conditions of PCR reaction, volume of reaction, concentration of magnesium ions, dNTPs, DNA polymerase type and concentration, additives as SYBR Green, DMSO etc. Name and manufacturer of qPCR instrument.

Justification of RT-PCR reference gene choice for example by bestKeeper software- length of reference PCR product

Software used to analyse RT-PCR results, statistical method.

Citation of method used to calculate RT-PCR results- for example Livak and Schmittgen 2001 or other.

Response: I rewrote this paragraph “RNA purity, concentration and integrity of each sample were examined by NanoDrop 2000c bioanalyzer (Thermo Fisher Scientific, USA), respectively. Then, RT was carried out using 1 μg total RNA and 200 U M-MLV Transcriptase (TaKaRa) in a 20 μl volume. The reaction was conducted at 70 °C for 10 min, 42 °C for 60 min and 70 °C for 15 min. The RT product was diluted 40 times with sterile water as template. The qPCR was conducted on the BIO-RAD CFX system in a 20 μl volume containing 4 μl diluted cDNA, 250 nM forward primer, 250 nM reverse primer, and 1×SYBR Premix Ex Taq II (TaKaRa) using the following conditions: 95 °C for 3 min, 40 cycles of 95 °C for 15 sec, 59 °C for 15 sec and 72 °C for 15 sec. Each sample duplicated three times. The 40S ribosomal protein S8 (T_00004056) was used as an internal reference because its expression level is relatively stable in different tissues. Melting curves were analyzed to verify the specificity using the Bio-Rad CFX Manage software. The relative expression levels were calculated using the 2-ΔΔCT method [3].

Other

Paragraph 2.5

Is the composed library build from the same quantity of RNA from roots, stems and leaves or the proportions between RNA from these sources are different? It should be stated in material and methods section.

Response:Thanks for the suggestion. We have stated in material and methods section 4.1. “The pooled RNA with integrity number more than seven was used for the following RNA-seq, small RNA and degradome library construction.”

Captcha and contents of tables S19 and S20 are exchanged.

Response:

Thanks for the suggestion. We have checked them.

Reviewer 2 Report

   The manuscript by Wu et al. is devoted to important problem of revealing network of gene expression and regulation in Canabis sativa L. It is very complex work; their results seem to be potentially important. However, there are several comments.

  1. The work includes wide genetic investigations; however, it seems to be descriptive. I suppose that Section “Discussion” should be strongly extended (now, Discussion is similar to Conclusions). Maybe, physiological roles of revealed ways should be discussed.
  2. Descriptions of methods of too brief. For example, see Sections 4.3 and 4.5. Even if these are standard methods, these descriptions are not enough to understanding these methods.
  3. A final schema, which will summarize results of the work, will be very important for understanding overall results of work. It can make manuscript more “friendly” for potential readers.

   Thus, manuscript seems to be interesting; however, revision is necessary. “Mechanistic” interpretations of results should be clearer.

Author Response

Comments for reviewer 2:

The manuscript by Wu et al. is devoted to important problem of revealing network of gene expression and regulation in Canabis sativa L. It is very complex work; their results seem to be potentially important. However, there are several comments.

  1. The work includes wide genetic investigations; however, it seems to be descriptive. I suppose that Section “Discussion” should be strongly extended (now, Discussion is similar to Conclusions). Maybe, physiological roles of revealed ways should be discussed.

Response: MiRNAs and lncRNAs functioned in multiple pathways. We focused on the top five KEGG enrichment pathways of miRNAs/lncRNAs targets. For miRNAs, it was reported that miR-362, targeted RecQ-mediated genome instability protein 1 [4]. miR-125b targeted ER alpha-1, 2-mannosidase and played a critical role in maintaining protein homeostasis in the mammalian secretory pathway [5]. While, KAS II, GPI ethanolamine phosphate transferase 3 subunit O and CCR4-NOT transcription complex subunit 4 potentially targeted by miRNAs were not reported. KAS II was involved in fatty acid elongation from 16:0-ACP to 18:0-ACP [6]. GPI ethanolamine phosphate transferase 3 subunit O functioned in GPI-anchor biosynthesis [7]. While the top five KEGG enrichment pathways targeted lncRNAs were not revealed previously. For example, 1,4-alpha-glucan branching enzyme played an significant role in the biosynthesis of branched polysaccharides, glycogen, and amylopectin [8]. All these results indicted the important regulation of miRNAs and lncRNAs in C. sativa.

  1. Descriptions of methods of too brief. For example, see Sections 4.3 and 4.5. Even if these are standard methods, these descriptions are not enough to understanding these methods.

Response: We have revised the methods part, including sections 4.3 and 4.5.

  1. A final schema, which will summarize results of the work, will be very important for understanding overall results of work. It can make manuscript more “friendly” for potential readers.

Response: We have added a Figure abstract in the supplementary file.

Round 2

Reviewer 1 Report

In my opinion Authors did their best to improve the manuscript, they corrected all suggested points. I do not have any other comments to their work.

Reviewer 2 Report

Authors considered my questions and comments. I have not other remarks.